Particulate matter 10 induces oxidative stress and apoptosis in rhesus macaques skin fibroblast

Lee Jiin 1 2
Kwon Jeongwoo 1
Jo Yu-Jin 1
http://orcid.org/0000-0002-0056-4995 Yoon Seung-Bin 1
Hyeon Jae-Hwan 1
Park Beom-Jin 1
You Hyeong-Ju 1
Youn Changsic 1
Kim Yejin 1
Choi Hyun Woo 2 choihw@jbnu.ac.kr
Kim Ji-Su 1 kimjs@kribb.re.kr
1 Primate Resources Center, Korea Research Institute of Bioscience and Biotechnology , Jeongup-si , Republic of Korea
2 Department of Animal Science, Jeonbuk National University , Jeonju, Jeollabuk-do , Republic of Korea
Wang Xinfeng
Electronic publication date: 2023 Dec 18
Publication date: 2023
Volume: 11
Electronic Location ID: e16589
Received 2023 Aug 2; Accepted 2023 Nov 14
Copyright: © 2023 Lee et al.
Copyright year: 2023
Copyright holder: Lee et al.
License: This is an open access article distributed under the terms of the Creative Commons Attribution License, which permits unrestricted use, distribution, reproduction and adaptation in any medium and for any purpose provided that it is properly attributed. For attribution, the original author(s), title, publication source (PeerJ) and either DOI or URL of the article must be cited.
License URL: https://creativecommons.org/licenses/by/4.0/

Keywords: Non-human primate, Particulate matter 10, Toxicity, Reactive oxidative stress, Apoptosis

Funding: Korea Research Institute of Bioscience and Biotechnology (KRIBB) KGM5162322 This research was supported by the Korea Research Institute of Bioscience and Biotechnology (KRIBB) Research Initiative Program KGM5162322. The funders had no role in study design, data collection and analysis, decision to publish, or preparation of the manuscript.

==============================
Background

Particulate matter (PM) is a major air pollutant that affects human health worldwide. PM can pass through the skin barrier, thus causing skin diseases such as heat rash, allergic reaction, infection, or inflammation. However, only a few studies have been conducted on the cytotoxic effects of PM exposure on large-scale animals. Therefore, herein, we investigated whether and how PM affects rhesus macaque skin fibroblasts.

Methods

Rhesus macaque skin fibroblasts were treated with various concentrations of PM10 (1, 5, 10, 50, and 100 μg/mL) and incubated for 24, 48, and 72 h. Then, cell viability assay, TUNEL assay, and qRT-PCR were performed on the treated cells. Further, the reactive oxygen species, glutathione, and cathepsin B levels were determined. The MTT assay revealed that PM10 (>50 μg/mL) proportionately reduced the cell proliferation rate.

Results

PM10 treatment increased TUNEL-positive cell numbers, following the pro-apoptosis-associated genes (CASP3 and BAX) and tumor suppressor gene TP53 were significantly upregulated. PM10 treatment induced reactive oxidative stress. Cathepsin B intensity was increased, whereas GSH intensity was decreased. The mRNA expression levels of antioxidant enzyme-related genes (CAT, GPX1 and GPX3) were significantly upregulated. Furthermore, PM10 reduced the mitochondrial membrane potential. The mRNA expression of mitochondrial complex genes, such as NDUFA1, NDUFA2, NDUFAC2, NDUFS4, and ATP5H were also significantly upregulated. In conclusion, these results showed that PM10 triggers apoptosis and mitochondrial damage, thus inducing ROS accumulation. These findings provide potential information on the cytotoxic effects of PM10 treatment and help to understand the mechanism of air pollution-induced skin diseases.

Introduction

Increasing worldwide air pollution, particularly particulate matter (PM), a complex lead, dust, dirt, and sand with water vapor and gases, is associated with various health problems (Phalen, 2004). PM is categorized into two groups based on particle size: PM10 for 2.5–10 µm and PM2.5 for <2.5 µm in aerodynamic diameter (Jung et al., 2017). PM2.5 negatively affects internal organs, such as the lung, heart, ovary, and brain (Du et al., 2016; Gai et al., 2017; Genc et al., 2012; Liu et al., 2017). Skin protects the internal organs and is directly exposed to the air. Exposure to airborne PM causes various skin problems and diseases. Connective tissue maintains the structural integrity of a mammalian body. Connective tissue comprises various extracellular matrices and fibroblasts that contribute to several immune and inflammatory responses.

PM comprises endotoxin, metals, and polycyclic aromatic hydrocarbons (PAHs) that exhibit biological toxicity. PAHs bound to PM surface cause mammalian health problems through reactive oxygen species (ROS) generation (Idowu et al., 2019; Tobiszewski & Namiesnik, 2012). Increased ROS levels due to PM exposure critically damage the respiratory and cardiovascular systems, other organs, and skin (Brook et al., 2010; Schraufnagel et al., 2019; Wu et al., 2018). Therefore, the mammalian body primarily defends PM through the skin barrier and immune response. The relationship between PM and epidemiology has been reported (Krutmann et al., 2014; Araviiskaia et al., 2019; Kim, Cho & Park, 2016); however, PM-induced cellular toxicity mechanisms in the skin remain unknown.

Fibroblasts, the most common cell type in mammals, secrete collagen, an important fibrous connective tissue component. Fibroblasts are majorly used in cell biology, disease, and toxicity research, owing to their association with tissue development and maintenance and disease (Kalluri, 2016; Lynch & Watt, 2018; Wynn, 2004). In addition, fibroblasts are involved in cell signaling and play a major role in various developmental processes, physiological functions, and diseases (Driskell & Watt, 2015). Fibroblasts are easily exposed to external contaminants, environmental toxins, and UV or ionizing radiation (Watanabe et al., 2004), and then cell-induced oxidative stress and inflammatory response cause extensive damage (Mittal et al., 2014). Concentrated ambient particles (CAPs) penetrate skin tissue and cause cell damage by inducing ROS generation and apoptosis (Magnani et al., 2016). However, the toxic effect of PM10 on fibroblasts and its pathophysiological mechanisms remain unclear.

The present study elucidated the cellular toxicity of PM on non-human primate primary fibroblasts. In addition, we determined whether PM-induced ROS and apoptotic-pathway activation promotes cell damage.

Materials and Methods

Reagents

All reagents were purchased from Sigma-Aldrich (St. Louis, MO, USA) unless stated otherwise.

Animals

All Rhesus monkeys were housed in the Primate Resources Center (Jeongup, South Korea) and were clinically asymptomatic for known diseases. Primary fibroblasts were isolated from rhesus macaques skin tissues. The experiments were reviewed and approved by the Institutional Animal Care and Use Committee (IACUC) of Korea Research Institute of Biotechnology & Biosciences (project number: KRIBB-IACUC-23009).

Cell culture and PM10 treatment

Rhesus macaque (animal #R0100, #R1055, Primate Resources Center (PRC), Korea Research Institute of Bioscience and Biotechnology (KRIBB)) skin fibroblasts were isolated from skin. Briefly, skin tissue was washed several times in Dulbecco’s phosphate-buffered saline (DPBS) with 1% penicillin–streptomycin (10,000 U/mL, Gibco, Carlsbad, CA, USA), cut into small pieces, and incubated with 0.25% trypsin solution at 30 min. After filtering through a sterile 70 μm cell strainer (Corning, Corning, NY, USA), cells were suspended using base medium composition with Dulbecco’s modified Eagle’s medium (DMEM; Gibco, Carlsbad, CA, USA) with 10% fetal bovine serum (FBS; Gibco, Carlsbad, CA, USA), 1% penicillin–streptomycin, non-essential amino acids (NEAA, Gibco, Carlsbad, CA, USA), glutamax (Gibco, Carlsbad, CA, USA), and 0.1% 2-mercaptoethanol at 37 °C and 5% CO2. Obtained primary skin fibroblasts were used for the PM10 treatment experiment within passage 9. PM10 (#ERM-CZ120; Sigma-Aldrich, Burlington, MA, USA) was dissolved in ultrapure water (Sigma, Burlington, MA, USA) and sonicated for 30 min with an ultrasonic homogenizer (Sonic Ruptor 400, OMNI, Kennesaw, GA, USA). Soluble PM10 was diluted with DMEM according to experiment concentrations (1, 5, 10, 50, and 100 µg/mL) at 4 °C at least 12 h. When rhesus fibroblasts were treated with PM10, PM10 solution for each concentration was filtered through a 0.22 μm pore size filter system to reflect the particle size penetrating the dermis. Finally, PM-dissolved media was added on attached fibroblasts with at 37 °C and 5% CO2. Detail component in PM10 is shown an Table S1.

Cell counting and viability assay (MTT assay)

Fibroblasts were seeded in six-well plates (3 × 105 cells/well) with a base medium. The fibroblasts were treated with PM10 of various PM concentrations (1, 5, 10, 50, and 100 µg/mL) and incubated for 72 h. To assess cell proliferation, the cells were incubated with a 0.05% trypsin solution for 3 min. Cells were suspended using base medium, and then mix 10 µL of cell with 10 µL of trypan blue, and pipet into a disposable countess chamber slide. Total cells were counted using the Countess™ 3 Cell Counter (Invitrogen, Waltham, MA, USA). As a result, To evaluate the cell viability, fibroblasts were seeded in triplicate in 96-well plates (5 × 103 cells/well) with a base medium. The fibroblasts were treated with PM10 of various PM concentrations (1, 5, 10, 50, and 100 μg/mL) and incubated for 24, 48, and 72 h. To determine cell viability, an MTT assay was performed using CyQUANT™ MTT Cell Viability Assay Kit (Invitrogen, Waltham, MA, USA) according to the manufacturer’s protocol. Absorbance was measured at 570 nm using a microplate spectrophotometer (Epoch 2; BioTek, Winooski, VT, USA).

TUNEL assay

Fibroblasts were seeded in 96-well plates (5 × 103 cells/well), treated with 50 μg/mL PM10, and fixed using 4% formalin solution at 24, 48, and 72 h. To detect apoptotic cells, the fixed cells were exposed to an in situ cell death detection kit (Roche, Basel, Switzerland) for 2 h and washed thrice with DPBS. The nuclei were stained with Hoechst 33342 (Sigma, Burlington, MA, USA), and apoptosis-positive cells were examined under the EVOS M7000 imaging system (Invitrogen, Carlsbad, CA, USA).

RNA isolation and quantitative real time–PCR

mRNA was isolated from the cell using RNeasy Micro Kit (Qiagen, Hilden, Germany) and reverse transcribed for cDNA synthesis in ReverTra Ace-αTM (Toyobo, Osaka, Japan) following the manufacturer’s instructions. Quantitative real time–PCR (qRT-PCR) was performed using cDNA and PowerUp SYBR Green Master Mix (Thermo Fisher Scientific, Waltham, MA, USA) on a StepOnePlus Real-Time PCR System (Thermo Fisher Scientific, Waltham, MA, USA) following the manufacturer’s protocol. Rhesus GAPDH was used as an internal control in all experiments, and other primer information is listed in Table S2. The 2–ΔΔCt method was used to analyze the qRT-PCR results in all experiments.

ROS, glutathione, and cathepsin B staining

To determine the levels of intracellular ROS, glutathione (GSH), and cathepsin B, cells were cultured overnight in 96-well black plates (5 × 103 cells/well). The media for the control and PM 50 μg/mL was changed and incubated at 37 °C and 5% CO2 for 72 h. The medium was removed and stained with DPBS containing 5 mM 2′,7′-dichlorodihydrofluorescein diacetate (DCFH-DA; Invitrogen, Carlsbad, CA, USA) at 37 °C for 20 min for intracellular ROS. After incubation, cells were washed and observed using a fluorescence microscope (EVOS M7000 imaging system). Cathepsin B was stained using Magic Red Cathepsin B Assay Kit (Immunochemistry Technologies LLC, Bloomington, MN, USA) and GSH was stained using 4-chloromethyl-6,8-difluoro-7-hydroxycoumarin (CMF2HC; Invitrogen, Carlsbad, CA, USA) according to the manufacturer’s protocol. The fluorescence intensity in the cytoplasm was quantified using Image J.

JC-1 assay and MitoSOX staining

Cells were seeded in 96-well black plates (5 × 103 cells/well) and incubated for 24 h. The media for the control and PM 50 μg/mL was changed for and incubated for 72 h. The cells were stained with 5,5′,6,6′-tetrachloro-1,1′,3,3′-tetraethylbenzimidazolylcarbocyanine iodide (JC-1; cat TA700, Life Technologies, Carlsbad, CA, USA) or 5 μM MitoSOX red (Life Technologies, Carlsbad, CA, US), respectively, at 37 °C for 20 min. After three times of washing with DPBS, the cells were observed using a fluorescence microscope (EVOS M7000 imaging system). The fluorescence intensity in the cytoplasm was quantified using Image J.

Data analysis

All statistical analyses and graph were performed using the Prism 8 software (Graphpad. Inc, La Jolla, CA, USA). The experiments were performed with three biological replicates. Significant differences in MTT assay were determined by one-way ANOVA, followed by Tukey’s multiple comparison tests. Another all experiments were determined by the t-test. The data were presented as mean ± standard error of the mean (SEM). p < 0.05 was considered statistically significant.

Results

Dose-dependent cytotoxicity induced by PM10 treatment in rhesus skin fibroblast

To investigated the cellular effect of environmental toxin PM10, we used the PM10 from European reference material ERM®-CZ120 and detail information was listed on Table S1. And, we observed the effect of PM10 on the cell viability of rhesus skin fibroblast. We treated the cells with 1, 5, 10, 50, or 100 µg/mL PM for 24, 48, and 72 h. Bright-field microscopy images showed that total cells were decreased dose-dependently of PM10 treatment compared with control (Fig. 1A). Additionally, a cell MTT assay and celll counting were performed to quantify cell proliferation and viability (Figs. 1B–1E). PM10 treatment did not affect the cell viability at 24 and 48 h. PM10 treatment at 72 h with a concentration exceeding 50 µg/mL significantly decreased fibroblast cell survival time-dependently. Therefore, we conducted the following experiments with 50 µg/mL PM10 and 72 h. To assess PM-induced cell death, we confirmed cellular apoptosis using TUNEL assay (Figs. 2A and 2B). Fibroblasts were treated with 50 µg/mL PM10 at 24, 48, and 72 h. PM did not affect cell apoptosis after 24 and 48 h of its treatment to fibroblasts. However, TUNEL-positive cells were significantly increased after PM10 treatment for 48 and 72 h. BAX, CASP3, TP53 and CYCS is major pro-apoptotic mediator after cellular damages and including the diseases (O’Brien & Kirby, 2008). Therefore, we confirm the mRNA expression level of these genes using qRT-PCR. The expressions level of BAX, CASP3, TP53, and CYCS were significantly increased after PM10 treatment to skin fibroblast (Fig. 2C). These results indicate that cell apoptosis is induced by PM10 treatment in rhesus skin fibroblast.

Figure 1 Effects of particulate matter 10 (PM10) on the viability of rhesus ear fibroblasts.

(A) Microscopic evaluation of cells. Ear fibroblasts were cultured at the indicated PM10 concentrations for 24, 48, and 72 h. (B–D) Percentage of cytotoxicity evaluated using the MTT assay each time. (E) Total cell numbers in PM10-treated groups at 72 h. Scale bar = 50 µm. Data are presented as the mean ± SEM. *p < 0.05, **p < 0.01, ***p < 0.001.

Figure 2 Effects of PM10 exposure on apoptotic cell.

(A) Apoptosis was evaluated by TUNEL staining of control and 50 µg/mL PM10-treated groups for 24, 48, and 72 h. (B) Percentage of TUNEL-positive control and PM10-treated cells at 24, 48, and 72 h. (C) Relative mRNA expression of the pro-apoptotic genes BAX, CASP3, TP53, and cytochrome C gene CYCS after 72 h for the indicated groups. Scale bar = 50 µm. Data are presented as the mean ± SEM. *p < 0.01, **p < 0.0001.

ROS generation with PM10 treatment

Oxidative stress occurs due to an imbalance between ROS levels and the antioxidant defense mechanism (Birben et al., 2012). Previous studies have shown that PM induces ROS accumulation, which leads to apoptosis through DNA and mitochondrial damage (Diao et al., 2021). To determine PM-induced cellular ROS levels in skin fibroblast, we quantified ROS production by fluorescent DCF reaction (the result of DCFH oxidation by different peroxides) in PM10-treated skin fibroblasts. DCF fluorescence gradient increased time-dependently in control and PM10 groups. Additionally, total ROS production was significantly enhanced in the PM10-treated groups compared with the control (Figs. 3A–3C). PM10-treated fibroblasts showed an increased cathepsin B intensity (Figs. 3D and 3E). GSH is the main intracellular antioxidant that maintains mitochondrial homeostasis (Mari et al., 2020). Representative glutathione, and antioxidant enzymes, such as catalase (CAT), glutathione peroxidase 1 and 3 (GPX1 and GPX3), are the main antioxidant-defense mechanisms in mammals (Chu et al., 2016). In the PM10-treated group, GSH levels were significantly decreased (Figs. 3F and 3G). Furthermore, mRNA expression levels of ROS and antioxidant enzyme-related genes (CAT, GPX1, and GPX3) were increased after PM10 treatment (Fig. 3H). These results indicated that PM10 promotes ROS generation through the changes in the expression of ROS-related genes.

Figure 3 PM10 exposure induces oxidant.

(A) ROS expression level was evaluated after 72 h of PM10 treatment. Scale bar = 50 µm. (B) The fluorescence intensity of ROS normalized at 72 h after treatment for the indicated groups. (C) The normalized relative intensity of ROS. (D) Catepsin B expression levels was measured by intensities of Catepsin B. (E) The normalized relative intensity of cathepsin B. (F) GSH expression level was showed that fluorescent microscopy, and (G) normalized relative intensity of GSH. (H) Relative mRNA expression of cytoplasmic ROS and antioxidant enzyme-related genes after 72 h of PM treatment. Data are presented as the mean ± SEM. *p < 0.01, **p < 0.0001.

PM10 treatment caused mitochondrial dysfunction in rhesus fibroblasts

To determine PM-induced mitochondrial effects, we tested the mitochondrial membrane potential (ΔΨm) by JC-1 staining. After JC-1 staining, we assumed the ratio of J-aggregates(red)/J-monomers (green) signal between control and PM10 treated fibroblasts (Figs. 4A and 4B). The PM10-treated groups displayed J aggregation. In mitochondrial ROS detection, PM10-treated groups showed increased MitoSOX intensities (Figs. 4C and 4D). Mitochondrial complexes (complex I–IV) are required for proper mitochondrial functioning. NADH (ubiquinone oxidoreductase) genes involved in the mitochondrial complexes control the overall mitochondrial function via proton translocation and ΔΨm. Mitochondrial complex proteins, such as NADH ubiquinone dehydrogenase-encoding genes (NDUFA2 and NDUFS4), NADH dehydrogenase-encoding genes located in mitochondrial complex 1 (NDUFA1 and NDUFC2) and the mitochondrial ATP synthase gene (ATP5H), has a function for maintaining the mitochondrial membrane potential and chemical proton gradient through oxidative phosphorylation via electron transport and ATP production (Zorova et al., 2018). In PM10-treated groups, NDUFA1, NDUFA2, NDUFC2, NDUFS4, and ATP5H mRNA levels were significantly decreased compared with the control (Fig. 4E). These results suggest that PM10 affects mitochondrial functions via mitochondrial complex protein regulation.

Figure 4 PM10 induced mitochondrial dysfunction in ear fibroblast.

(A) JC-1 staining to represent mitochondrial membrane depolarization. Cells were strained after 72 h of PM10 exposure. Scale bar = 50 µm. (B) Mitochondrial membrane depolarization levels were represented by measuring ∆Ψm (red/green). (C) Mitochondrion-specific superoxide was detected by MitoSOX red staining. (D) Relative MitoSOX fluorescent intensity. (E) Relative mRNA expression of mitochondrial genes NDUFA1, NDUFA2, NDUFAC2, NDUFS4, and ATP5H after 72 h of PM treatment. Data are presented as the mean ± SEM. *p < 0.01, **p < 0.0001.

Discussion

In this study, we demonstrated the cytotoxic effects, such as apoptosis, ROS generation through related gene expression regulation and mitochondrial damage by treatment of PM10 on rhesus skin fibroblasts (Fig. 5). The results revealed that a high PM10 dose reduced cell proliferation and increased apoptotic cell numbers by regulating apoptosis-related gene expression. Furthermore, PM10-treated fibroblasts showed increased ROS levels and decreased ΔΨm. Our data may help us understand the causes cellular damage by cutaneous exposure to PM10.

Figure 5 A simple model for PM-induced fibroblast cellular toxicity.

PM10 can penetrate the dermis layer through the skin barrier and affects dermal fibroblasts directly and indirectly. PM10 exposure induces apoptosis by upregulating apoptotic gene expressions. Mitochondrial damage and oxidative stress-related gene regulation increased cellular ROS and mitoSOX levels. The activation of GSH and cathepsin B were induced by PM10 exposure. The environmental pollutant PM10 triggers cellular toxicity in fibroblasts.

PM can cause skin diseases, such as atopic dermatitis, acne, skin aging, and cancer (Kee et al., 2021; Kim, Cho & Park, 2016; Song et al., 2011; Vierkotter et al., 2010). PM can penetrate the epidermis and intercellular space through the stratum spinosum with existing hair skin. PM can break the skin barrier function by inhibiting filaggrin (FLG) protein expression in humans and murine epithelial cells (Kim et al., 2021; Lee et al., 2016) and affect the epidermal and dermal environment, including fibroblasts. Furthermore, PAHs cross epithelial barriers from the pulmonary alveoli and enter systemic circulation, providing a potential alternate route to reach the deeper skin layers (Ruchirawat et al., 2007). These findings demonstrated that exposure of PM can affect the deeper layer cell environment directly. Our results also show that PM10 cause the cellular toxicity by reducing the proliferation rate and inducing the cell apoptosis (Fig. 1). Contrary to our studies, PM 10 treatment did not affect cell viability in nasal polyp-derived fibroblasts (NPDFs) while increase the IL-33/ST2 pathway-mediated immune response (Lee & Kim, 2022). We speculate that this discrepancy may be due to the period of PM treatment (24H–72H), different tissue (skin and nasal polyp) and species (human and rhesus monkey). However, this results also demonstrated that PM10 is potential mediator could lead to disease exacerbation or progression by affecting molecular mechanism changes.

Previous studies show that PM cause the apoptosis in skin cells (Jun et al., 2020; Piao et al., 2018). Our results showed that PM10-treated fibroblasts displayed reduced cell proliferation rates and cell apoptosis, while the apoptosis-related gene (BAX and Bcl-2) expression was upregulated. Moreover, TP53, a tumor suppressor protein that repairs DNA, was also increased. PM exposure increases lung cancer mortality in humans but is not primarily associated with skin cancer (Coleman et al., 2020; Pope et al., 2020); however, PM10 causes TP53-dependent cell death in various cell types (Jo et al., 2020; Soberanes et al., 2006). PM10 induced cell apoptosis and reduced cell proliferation via the regulation of apoptosis-related genes. Therefore, the results indicate that the toxic effects of PM10 on rhesus fibroblast are the reason behind the regulation of cellular damage markers, leading to increased apoptosis and decreased cell proliferation.

Active ROS are involved in several cellular processes and cell survival, such as increasing cell damage, oxidative stress, and DNA damage. Therefore, ROS must be studied to enhance the survival of all aerobic organisms. Many defense mechanisms exist for cell maintenance and survival between ROS production and removal. An imbalance in oxidation-promoting states is often referred to as oxidation stress. In skin tissue, PM infiltration into the dermis increases intracellular ROS levels, leading to aryl hydrocarbon receptor (AHR)-mediated cellular damage (Araviiskaia et al., 2019; Kim et al., 2017; Vogeley et al., 2019). Additionally, PM causes skin barrier dysfunction and increases ROS levels through toll-like receptor (TLR)-mediated inflammation signaling (Ryu et al., 2019). In our study, fibroblast, the main cell layer forming the dermis, showed increased ROS levels after PM treatment (Figs. 3A and 3B). High ROS levels impair the lysosome damage and cathepsin B release in fibroblasts (Song & Hwang, 2020). Increased cathepsin B activity induces IL-1β production, leading to pyroptosis with subsequent caspase-1 activation (Chevriaux et al., 2020). Additionally, PM activates the NLRP3 inflammasome, thereby releasing cathepsin B into the cytosol. Our results showed that cathepsin B intensity in fibroblast cytoplasm was increased after PM treatment (Figs. 3C and 3D). These results demonstrated that PM-induced oxidative stress induces cathepsin B activity by pyroptosis. Further study is required on the relationship between PM-induced NLRP3 inflammasome activation and cellular toxicity in skin fibroblasts. ROS cause changes in cell survival or apoptotic mechanisms, including mitochondrial damage (Chaudhary et al., 2016). Mitochondrial damage can be detected using the membrane-permeant dye JC-1 to monitor the status of mitochondrial membrane polarization (healthy state) or depolarization (damaged state) revealed by the red/green fluorescence intensity ratio (Lee et al., 2017). JC-1 red signal was decreased in PM10 treated groups (Figs. 4A and 4B), indicating that PM10 depolarized the mitochondrial membrane. PM can alter mitochondrial membrane polarization by inducing ROS production (Kyung et al., 2012; Lee et al., 2021). In mitochondrial damage by PM, NADH dehydrogenase gene expression levels were decreased dynamically following mitochondrial dysfunction (Li et al., 2017). Mitochondrial complex proteins, such as NDFUA1, NDUFA2, NDUFC2, NDUFS4, and ATP5H, located in the mitochondrial inner membrane, maintain the mitochondrial membrane potential (Δψm) and chemical proton gradient (ΔpH) through oxidative phosphorylation via electron transport and ATP production (Zorova et al., 2018). PM10 caused the downregulation of NDFUA1, NDUFA2, NDUFC2, NDUFS4, and ATP5H genes (Fig. 4E) and reduced the ROS level. These results indicate the general mitochondrial damage through the regulation of intracellular ROS production.

Conclusions

In this study, we revealed the relationship between particulate matter and increased cellular apoptosis. PM10-treated rhesus fibroblasts showed mitochondrial damage and ROS production. These findings contribute to understanding the cytotoxic effects of PM on dermal fibroblasts.

Supplemental Information

Supplemental Information 1 Supplementary tables.

Click here for additional data file.

Supplemental Information 2 Raw data in Figures 1, 2, 3, and 4.

Click here for additional data file.

Additional Information and Declarations

Competing Interests

Author Contributions

Data Availability

The authors declare that they have no competing interests.

Jiin Lee conceived and designed the experiments, performed the experiments, analyzed the data, prepared figures and/or tables, authored or reviewed drafts of the article, and approved the final draft.

Jeongwoo Kwon conceived and designed the experiments, performed the experiments, analyzed the data, prepared figures and/or tables, authored or reviewed drafts of the article, and approved the final draft.

Yu-Jin Jo conceived and designed the experiments, performed the experiments, analyzed the data, prepared figures and/or tables, and approved the final draft.

Seung-Bin Yoon performed the experiments, analyzed the data, prepared figures and/or tables, and approved the final draft.

Jae-Hwan Hyeon performed the experiments, analyzed the data, prepared figures and/or tables, and approved the final draft.

Beom-Jin Park performed the experiments, analyzed the data, prepared figures and/or tables, and approved the final draft.

Hyeong-Ju You performed the experiments, analyzed the data, prepared figures and/or tables, and approved the final draft.

Changsic Youn performed the experiments, analyzed the data, prepared figures and/or tables, and approved the final draft.

Yejin Kim performed the experiments, analyzed the data, prepared figures and/or tables, and approved the final draft.

Hyun Woo Choi conceived and designed the experiments, prepared figures and/or tables, authored or reviewed drafts of the article, and approved the final draft.

Ji-Su Kim conceived and designed the experiments, analyzed the data, prepared figures and/or tables, authored or reviewed drafts of the article, and approved the final draft.

The following information was supplied regarding data availability:

The raw measurements are available in the Supplemental Files.

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
