# Peer review of "Particulate matter 10 induces oxidative stress and apoptosis in rhesus macaques skin fibroblast"

_PeerJ, doi:10.7717/peerj.16589_

## Round 0.1 · original submission · Major Revisions

Based on the comments from the three anonymous reviewers, major revisions are required to further improve your manuscript.

Reviewer 1 ·

Basic reporting

This study investigates the cytotoxic effects of PM10 by using dermal fibroblasts obtained from Rhesus macaques. The findings demonstrate that PM10 leads to mitochondrial dysfunction and an increase in reactive oxygen species (ROS), which are identified as the primary cause of PM10 cytotoxicity. Considering the growing concern about the potential toxicity of air-derived PM10, the toxicity observed in skin fibroblasts in this study holds significant meaning.

Experimental design

This manuscript is well design for discover the effect of PM 10 in skin fibroblasts cell.

Validity of the findings

No comment

Additional comments

No comment

Reviewer 2 ·

Basic reporting

1. In Line 36, RT-qPCR is inconsistent with the qRT-PCR stated in line 129.
2. In Line 155, it contains two instances of "at".
3. "p" is not in italics in line 164.
4. In Line 180, "Fig 2" is missing "E" and "H".
5. I and J is not included in Figure 3 in Line 191.
6. The mRNA expression of CAT, GPX1, and GPX3 in Line 191 and Line 203 is not italicized.

Experimental design

1. It should be clearly stated the details of PM10. Additionally, the characterization of PM10 should be elaborated upon in the results section.
2. What is the proportion of fibroblasts in the skin tissue? In Line 169, does the exposure concentration of PM10 in experiment comply with the environmental exposure level?
3. In Line 179, the relationship between the genes (BAX, CASP3, TP53, and CYCS) and apoptosis needs to be explained.

Validity of the findings

1. In Line67,the expression of “the relationship between PM and epidemiology has been reported”is inaccurate. And authors should provided referemces.
2. In line170,how to ensure objectivity in photography? And I cannot infer the conclusions summarized in the article from images. For example, it's impossible to determine the dose-response effect of PM10 exposure on cell count after 24-hours exposure. The conclusions are inaccurate.
3. In Line179, there is no explanation of the relationship between ROS and GSH, as well as the gene expression of CAT, GPX1, GPX3. Without a clear understanding of the relationship between these factors, it is not possible to derive conclusions from this section."

Additional comments

Merely observing changes in gene expression and mitochondrial damage cannot conclusively prove that ROS is the causative factor. If mitochondrial damage or gene regulation is believed to be involved, further molecular biology experiments should be conducted for validation. For instance, this could involve the use of inhibitors or activators to demonstrate the role of mitochondria or the regulatory impact of related genes on ROS production.

Reviewer 3 ·

Basic reporting

The article is interesting, and there are references that may be useful to enrich the discussion.

Effect of Airborne Particulate Matter on the Immunologic Characteristics of Chronic Rhinosinusitis with Nasal Polyps - https://doi.org/10.3390/ijms23031018

Particulate Matters Induce Apoptosis in Human Hair Follicular Keratinocytes - 10.5021/ad.2020.32.5.388

This work specifically evaluates the direct cellular action of PM10.

And it is very important to understand the cellular response induced by this toxicant, as well as to relate it to what already exists in the literature. But organization of data and hypotheses requires attention.

The discussion is confusing. You point at line 214 that your experiments may help to understand allergic reactions, what results?

Linne 221 to 223 the filtration process belongs to the Material and Method section

The conection between the ROS, mitochondrial assays and the choosen genes are not clear yeat.

I suggest leading the text first through the direct damage done to cells, and how this damage relates to modifying gene activation/inactivation discussing the possible events that result in the observed activity

Experimental design

Good

Validity of the findings

Good

Additional comments

None

---

## Round 0.2 · accepted · Accept

The authors have addressed all of the reviewers' comments and have been evaluated by the reviewer and editor. One reviewer with major comments did not return the final comment on time, but we cannot wait any longer. The manuscript is ready for publication.

Reviewer 2 ·

Basic reporting

No comment

Experimental design

No comment

Validity of the findings

No comment